# An Audio-Based SLAM for Indoor Environments: A Robotic Mixed Reality Presentation

**DOI:** 10.3390/s24092796

**Published:** 2024-04-27

**Authors:** Elfituri S. F. Lahemer, Ahmad Rad

**Affiliations:** Autonomous and Intelligent Systems Laboratory, School of Mechatronic Systems Engineering, Simon Fraser University, Surrey, BC V3T 0A3, Canada; arad@sfu.ca

**Keywords:** audio-based SLAM, landmarks, EKF/ellipsoidal landmark-based SLAM, robotic mixed reality, Microsoft HoloLens, landmarks, HoloSLAM, Nao humanoid robot

## Abstract

In this paper, we present a novel approach referred to as the audio-based virtual landmark-based HoloSLAM. This innovative method leverages a single sound source and microphone arrays to estimate the voice-printed speaker’s direction. The system allows an autonomous robot equipped with a single microphone array to navigate within indoor environments, interact with specific sound sources, and simultaneously determine its own location while mapping the environment. The proposed method does not require multiple audio sources in the environment nor sensor fusion to extract pertinent information and make accurate sound source estimations. Furthermore, the approach incorporates Robotic Mixed Reality using Microsoft HoloLens to superimpose landmarks, effectively mitigating the audio landmark-related issues of conventional audio-based landmark SLAM, particularly in situations where audio landmarks cannot be discerned, are limited in number, or are completely missing. The paper also evaluates an active speaker detection method, demonstrating its ability to achieve high accuracy in scenarios where audio data are the sole input. Real-time experiments validate the effectiveness of this method, emphasizing its precision and comprehensive mapping capabilities. The results of these experiments showcase the accuracy and efficiency of the proposed system, surpassing the constraints associated with traditional audio-based SLAM techniques, ultimately leading to a more detailed and precise mapping of the robot’s surroundings.

## 1. Introduction

Advancements in artificial intelligence (AI) have facilitated the transition to a new era of versatile, efficient, and affordable autonomous robots [1]. They are employed in various indoor and outdoor tasks such as mapping, localization, pathfinding, obstacle avoidance, guiding, guarding, and providing care for the elderly, and most notably, autonomous navigation [2]. Autonomous navigation is crucial in many applications as it enables a robot to safely and effectively traverse complex, unstructured environments. The robot must be able to simultaneously build a map of its surroundings and determine its location within that map [3,4]. This process is now well established and is referred to as Simultaneous Localization and Mapping (SLAM) [5]. The SLAM problem can be applied to a wide variety of environments, including both static and dynamic, indoor and outdoor, with different robotic platforms such as ground robots, underwater robots, and aerial drones [5,6,7]. Considerable effort has been dedicated to crafting efficient solutions for the SLAM problem [8,9]. In its early stages, SLAM primarily relied on range sensors, such as sonar, lasers, and cameras, as the principal information sources for constructing maps and ascertaining the robot’s position and orientation [10]. Within indoor environments, the majority of SLAM and navigation systems depend on visual data. Vision imparts a wide range of capabilities to robots, rendering cameras a universally integrative component [11]. However, vision-based SLAM faces limitations like a restricted field of view and occlusion, hampering target exploration. Ground conditions heavily affect its accuracy [12]. Sensor fusion and advanced algorithms, like machine and deep learning, enhance mapping accuracy and resilience against limitations and changes [13]. An alternative approach is to integrate audio input into the robot’s navigation system, thereby broadening its sensory capabilities and facilitating navigation in scenarios where visual data may be inadequate [14,15,16,17]. Indeed, an auditory system enables the robot to comprehend spoken instructions, identify particular sounds or speakers, determine the source of sounds accurately, and react to important environmental auditory cues. This includes sound source localization, speaker tracking, speech separation, recognition, and audio-based SLAM [18,19,20]. Computational auditory scene analysis (CASA) has progressed in understanding environmental sounds, focusing on source localization and separation [16,21,22]. In contrast, audio-based SLAM algorithms are relatively less mature and face certain challenges that may hinder their widespread adoption in robotics [23,24]. Audio-based SLAM typically involves several important steps. Firstly, data are acquired using one or more microphones, capturing sound waves from the environment. Preprocessing enhances data quality by filtering noise. Features are then extracted for localization and mapping. Sound source position is estimated using techniques like beamforming, triangulation, or time-delay estimation, along with combinations of signal processing methods, sensor fusion, and potentially machine learning approaches. Data association maintains the correspondence between observed sound sources and landmarks. State estimation integrates information from various sensors, and optimization refines trajectory and map estimates, minimizing errors [25].

When equipped with a microphone array, a robot can estimate sound source directions, but accurately gauging distance presents challenges, particularly when the distance surpasses the array’s dimensions. As a result, deducing Cartesian source positions from Direction of Arrival (DoA) estimates presents an issue with multiple unknowns. Additionally, the presence of reverberation and noise introduces errors in estimation that may result in incorrect source position estimations. Moreover, instances of silence, such as during human speech pauses, can result in the absence of audio source estimations. Consequently, research in sound source localization and SLAM for mobile robots has mainly focused on detecting the sound sources’ directions. Numerous theories and methods exist for microphone-array-based sound source localization, including Received Signal Strength (RSS), Angle of Arrival (AOA), Time of Arrival (TOA), Time Difference of Arrival (TDOA), Frequency Difference of Arrival (FDOA), Multiple Signal Classification (MUSIC), beamforming, and other advanced techniques [26]. In the TDOA algorithm, accurately estimating the sound source location hinges on effectively gauging the time difference of signals received by microphones [27]. This is achieved through two main approaches: cross-correlation methods like Generalized Cross-Correlation (GCC) and cross-power spectrum phase, and obtaining TDOA estimation via path impulse response calculations. The GCC with Phase Transform (GCC-PHAT) stands out as a specific approach frequently utilized for various sound localization tasks [19,28,29].

Current audio-based SLAM methods [30,31,32,33] typically assume open spaces and clear paths to multiple sound sources. Real-world scenarios, however, often feature reflective surfaces like narrow hallways, causing localization challenges. In addition, these audio-based SLAM solutions incorporate either artificial or natural audio sources as landmarks and these landmarks are progressively integrated into the robot’s map over time. Existing audio-based SLAM solutions often overlook the presence of landmarks due to the unavailability of direct paths to sound sources caused by reflections or detection issues. This limitation hampers the comprehensive mapping and localization of environments, urging the need for advancements in techniques to address such complexities effectively [34]. In an audio-based SLAM, landmarks can be extracted from audio signals or use audio sources themselves. In environments with multiple audio sources, localization ambiguity increases due to overlapping signals, reflections, and reverberations. Mapping becomes complex as each source contributes to the acoustic map. Identifying and tracking multiple sources requires advanced signal-processing techniques like source separation and clustering. Techniques like triangulation improve location estimation. However, managing multiple sources increases computational demands and system costs. Conversely, single-source audio SLAM offers simplicity and reduced complexity, albeit with potential localization and mapping inaccuracies. Identifying and tracking the main audio signal in single-source SLAM is crucial, yet poses challenges in complex environments with signal distortion and multiple sources. Our method utilizes single-source audio-based SLAM. For localization, we exclusively estimate the direction of the primary signal (active speaker) using the GCC-PHAT technique with a microphone array.

Mixed Reality (MR) offers a promising avenue to overcome the limitations of traditional landmark-based SLAM systems. By overlaying digital information onto physical surroundings, MR expands the perceptual capabilities of robots beyond tangible landmarks, allowing for more versatile mapping and localization. In MR-enhanced SLAM systems, virtual landmarks can be dynamically generated and manipulated, offering flexibility in adapting to diverse environments. These virtual landmarks may include not only visual cues but also auditory or spatial markers, aligning with the capabilities of audio-based SLAM methods. Additionally, MR and holographic displays facilitate the creation of interactive and immersive experiences, enabling robots to interact with both physical and virtual elements for enhanced localization and mapping accuracy [35]. Integrating Microsoft HoloLens or any other mixed-reality device [36,37] with a robot’s real-world environment enables the robot to effectively follow, track, communicate, and interact with specific speakers. Simultaneously, it empowers the robot to conduct a virtual audio-based SLAM with high accuracy and success, revolutionizing its ability to navigate and perceive its surroundings in dynamic and complex environments.

The main contribution of this paper over and above the state of the art is the integration of a microphone array platform, mixed-reality, and holographic displays on Microsoft HoloLens [37] to perform audio landmark-based SLAM in indoor environments. The proposed system is verified on a Nao robot [38] platform. The system begins with identifying a specific speaker in a multi-audio environment and extracting sound source information using the microphone array. Simultaneously, it employs a Short-Time Fourier Transform (STFT) to transform input signals into the complex domain and extract features using a combination of GFCC (Gammatone Frequency Cepstral Coefficients) and MFCC (Mel-frequency Cepstral Coefficients) [39]. These extracted features are then fed into multiple speaker classifiers, including Gaussian Mixture Model (GMM) [40], Support Vector Machine (SVM) [41], Convolutional Neural Network (CNN) [42], Deep Neural Network (DNN), and Time-Delay Neural Network (TDNN) [43], for keyword detection and sound source identification, including the source’s angle. This angle information is then used to position virtual landmarks in the robot’s environment via Microsoft HoloLens holographic apps. The research also involves the application of a traditional ellipsoidal SLAM algorithm to estimate the robot’s path and integrate virtual landmarks in the mapped environment. The robot is then able to navigate toward the specified speaker while avoiding interfering sound sources. This study demonstrates the effective localization of both the robot and sound sources in indoor environments, which has implications for improving robot navigation and interaction in real-world scenarios.

The paper’s organization is as follows: Section 2 provides a comprehensive review of the relevant literature, highlighting the present state of audio-based SLAM and its primary challenges. Section 3 elucidates the intricate design of our proposed system. Section 4 integrates simulation studies and an extensive exploration of the benefits inherent in our architecture. The paper culminates with a conclusion in Section 5. This structured approach guides readers through the background, system details, empirical assessments, and final insights of our research.

## 2. Related Studies

The idea of robot audition was first reported by Nakadai et al. [44]. Subsequently, researchers have explored numerous approaches to enhance sound source localization (SSL) for various applications in robotics [45,46,47]. The use of SSL in robotics is relatively new, dating back to 1989 when Squirt, the first robot equipped with an SSL module, was introduced [48]. After the Squirt robot was equipped with the ability to locate surrounding sound sources in 1989, the SSL field has continuously advanced to address challenges. In 1995, MIT’s Robert installed a basic robot auditory system. In 2006, the Honda Research Institute pioneered real-time tracking with IRMA and a robot-head microphone array integration [49]. These approaches involve collecting acoustic data from sound sources such as microphone arrays and integrating them with other sensory data such as vision and odometry information [50]. Filtering techniques [51,52] are then applied to leverage sound information alongside robot movement data to accurately estimate their position and orientation, which can be valuable for tasks such as navigation, mapping, and interaction with their environment. Conventional audio-based SLAM approaches primarily integrate SSL with SLAM [53]. These methods typically initiate TDOA estimation using multi-channel audio data from microphone arrays. Subsequently, the relative distances or angles between sound sources are then computed to assist SLAM implementation. In [33], a collection of sound sources served as landmarks, and a microphone array was mounted on a wheeled robot. This setup was designed for the concurrent localization of both the sound sources and the robot. Meng et al. [29] introduced an approach utilizing a microphone array combined with Light Detection and Ranging (LiDAR). The study successfully located the robot and mapped its environment in experiments. Nonetheless, to achieve satisfactory outcomes, precise motion data from odometry or LiDAR were required. The robot’s motion and its performance were hindered due to signal sync, noise, and DOA errors from indoor acoustics. Inaccurate motion reports limit effectiveness. Some SSL methods presented in [54,55,56] combined audio and visual data, focusing solely on visible sound sources, and are unsuitable for robot navigation when targets are hidden. Sasaaki et al. [57] designed a mobile robot with a microphone array for estimating multiple sound source positions by triangulating observations from various robot positions. Echoes [58] and multipath [59] have also previously been employed for SLAM and, more broadly, for estimating room geometry [60]. In [61], a method was proposed for localizing a mobile robot using structured sound sources that emit unique codes, similar to the GPS system, where the exact positions of each sound source are known beforehand. This is different from the SLAM approach where landmark locations are not known a priori. While effective for static sound sources, these methods struggle to adapt when the robot moves amidst dynamic sound changes and to accurately estimate the distance between the sources and the robots in such situations.

In contrast, the system introduced in this paper is initiated by identifying unique sound targets via pre-registered voiceprints. The angle of the target speaker is all that is required; the algorithm tracks the sound of interest, facilitating navigation and SLAM task execution with a virtual map. The estimated direction serves as observation data, and this became a standard bearing-only SLAM problem solely for guiding the robot to track the active speaker while localizing itself and creating a map of its environment. However, since there is a lack of additional information to perform a complete SLAM operation, the Ellipsoidal HoloSLAM algorithm [62] is employed. Ellipsoidal HoloSLAM addresses this problem by incorporating virtual landmarks into the mapping process, allowing for an accurate and realistic SLAM implementation without a need for an active predefined sound source locationprior to location. As the robot moves and the active speaker’s location and direction change, the SSL and SLAM algorithms work together to continually update the robot’s position and orientation within the built map, allowing it to follow the speaker and build a detailed virtual map of the environment at the same time. This approach has potential in indoor applications in areas such as human––robot interaction, assistive robotics, and indoor navigation.

## 3. Materials and Method

### 3.1. Overall System Architecture

The inference part of the proposed system was implemented on a Nao robot for real-time operation [63] as illustrated in Figure 1. The robot is equipped with a microphone circular array module, specifically the ReSpeaker microphone array module (Seeed Technology Co., Ltd., Shenzhen, China), on its head. The ReSpeaker module is connected to a Raspberry Pi 4B (Raspberry Foundation, Cambridge, UK), which is used to collect and process the recorded acoustic data [64].

Upon detecting a keyword, the ReSpeaker microphone array records acoustic data, feeding them into various classifier models (GMM, SVM, DNN, CNN, TDNN) to discern the active speaker’s angle. This allows the robot to adjust its position, promoting natural interaction and movement tracking. This has applications in human-robot dialogue, social robotics, and guided robot tours.

Subsequently, the robot employs the virtual Ellipsoidal HoloSLAM technique to map its surroundings and establish its position within the map. Unlike traditional acoustic-based SLAM, this approach leverages Microsoft HoloLens and mixed-reality technology to virtually map the environment. Virtual landmarks are incorporated into the robot’s surroundings while it moves, tracking the active speaker and efficiently avoiding obstacles.

### 3.2. Active Speaker Localization Using Microphone Array

This section focuses on Sound Source Localization (SSL), which involves identifying the direction and distance of detected sounds using electronic receivers, like microphones. We use the term “Sound Localization” for sound direction estimation. SSL system design is an active area of research, employing techniques like beamforming, cross-correlation, time delay estimation, and machine learning [65,66]. Figure 2 outlines the key stages of sound source direction estimation.

Upon confirming a received signal as a signal of interest, data from microphones undergo processing, including band-pass filtering. Time Difference Of Arrival (TDOA) between microphones is measured to determine the sound source location, often using the Generalized Cross-Correlation with Phase Transform weighting function (GCC-PHAT) method [67] (Figure 3).

The cross-correlation between two discrete signals $x_{1}\left(t\right)\;and\;x_{2}\left(t\right)$ received from the left and right channels can be defined by [29]:(1)$$R_{x_{1}x_{2}}\left(k\right)=\sum_{n=-\infty }^{\infty }x_{1}\left(n\right)\cdot x_{2}\left(n+k\right)$$

The cross-correlation can also be represented with the help of the convolution operator as follows:(2)$$R_{x_{1}x_{2}}\left(k\right)=x_{1}\left(-k\right)*x_{2}\left(k\right)$$

In practice, a limited signal segment is processed, estimating cross-correlation. The equation applies to two signals of length ***N***:(3)$$R_{x_{1}x_{2}}\left(k\right)=\frac{1}{N}\sum_{n=0}^{N-1-K}x_{1}\left(n\right)\cdot x_{2}\left(n+k\right)$$

For longer signals, Fourier transformation simplifies calculations, enabling frequency domain multiplication. The spectral cross-power density is defined as the Fourier transform of the cross-correlation function by:(4)$$S_{X_{1}{xX}_{2}}\left(f\right)=X_{1}\left(f\right)\cdot X_{2}^{*}\left(f\right)$$

Complex conjugation is denoted by (∙). Cross-correlation is calculated via inverse Fourier transformation. When one signal is a time-shifted version of another, cross-correlation has a peak at time *D*. The delay is expressed as:(5)$$\hat{D}=\arg\underset{k}{\max}{\hat{R}}_{x_{1}x_{2}}\left(k\right)$$

Real-time factors affect the maximum position. To enhance stability, Generalized Cross-Correlation (GCC) in [68] uses weight functions on cross-power spectral density. The general GCC equation is:(6)$${\hat{R}}_{x_{1}x_{2}}^{\left(g\right)}\left(\tau \right)=F^{-1}\left\{X_{1}\left(f\right)\cdot X_{2}^{*}\left(f\right)\cdot \psi (f)\right\}$$
where ***ψ*** stands for a weighting function.

Various weighting functions are available for GCC to improve time delay sensitivity. PHAT weight, as introduced in [69], can be defined as:(7)$$\psi_{p}\left(f\right)=\frac{1}{\left|X_{1}(f)\cdot X_{2}^{*}\left(f\right)\right|}=\frac{1}{\left|S_{X_{1}X_{2}}(f)\right|}$$

Inserted into Equation (6), the GCC-PHAT results in:(8)$${\hat{R}}_{x_{1}x_{2}}^{\left(p\right)}\left(\tau \right)=F^{-1}\left\{\frac{X_{1}(f)\cdot X_{2}^{*}\left(f\right)}{\left|X_{1}(f)\cdot X_{2}^{*}\left(f\right)\right|}\right\}=F^{-1}\left\{\frac{S_{X_{1}X_{2}}(f)}{\left|S_{X_{1}X_{2}}(f)\right|}\right\}$$

The position of the maximum ${\hat{R}}_{x_{1}x_{2}}^{\left(p\right)}\left(\tau \right)$ of corresponds to the delay between the signals:(9)$${\hat{D}}_{p}=\arg\underset{k}{\max}{\hat{R}}_{x_{1}x_{2}}\left(\tau \right)$$

For discrete signals, ${\hat{D}}_{p}$ represents time units through signal sampling frequency. Shift replaces delay. In acoustic source localization, a microphone pair, known as an “active” pair, is utilized to estimate sound source direction. The choice of this pair varies based on the array geometry and sound source direction. Typically, it consists of the two closest or the pair with the greatest time delay difference [68]. The calculation of the angle based on the signal propagation time difference between two microphone signals takes place here using the following equation:(10)$$\theta ={sin}^{-1}\left(\frac{\tau }{\tau_{max}}\right)$$

The ***τ*** is the signal runtime difference and $\tau_{max}$ is the maximum runtime between two microphones. $\tau_{max}$ can be calculated by the following:(11)$$\tau_{max}=\frac{d}{c}$$
where ***d*** is the distance between the two microphones and ***c*** is the speed of sound.

By substitute $\tau_{max}$ from Equation (11) into Equation (10), the angle based on the signal propagation time difference can be calculated as:(12)$$\theta ={sin}^{-1}\left(\frac{c\;\cdot \;\tau }{d}\right)$$

Using the plane wave model, the distance required for the wavefront to pass through both microphones can calculated as shown in Figure 4. The time difference can be calculated as:(13)$$\tau =\frac{d\;\sin(\theta )}{c}$$

An angle of **0°** signifies a wavefront perpendicular to the microphone axis, while **±90°** indicates a wavefront aligned with the microphone axis. When the signal propagation time difference *τ* is zero, the angle *θ* is 0°. At ***τ* =** $\tau_{max}$, *θ* is 90°, and at ***τ* =** $-\tau_{max}$, *θ* is −90°. For ***τ* ≠ ±**$\tau_{max}$, two angles are observed on the full circle, mirroring each other along the microphone axis. This means only angles between −90° and 90° are determinable in a linear microphone array, without information about whether the wavefront is above or below the microphone axis.

### 3.3. Audio-Based Ellipsoidal Virtual HoloSLAM Algorithm Implementation

In robotics, auditory systems are vital for human interactions and navigation tasks. Current research addresses multiple aspects, such as speaker localization, speech separation, enhancement, recognition, and speaker identification [65,70,71]. Speaker localization using biological hearing principles or microphone arrays has been a long-standing focus. Sound source localization (SSL) aims to automatically locate sound sources, which is crucial for a robot’s self-localization and mapping. The research objective here is to enable a robot to autonomously determine its location and map its surroundings while in motion, even without prior sound source knowledge. **Localization** here solely refers to estimating the robot’s position over time in a global frame, without prior knowledge of natural or artificial sound source landmarks.

HoloSLAM revolutionizes the landmark-based SLAM in autonomous robot navigation by merging the real and virtual worlds using Microsoft HoloLens and mixed-reality techniques [62]. It combines established methods to provide real-time environment construction and robot position tracking. Mixed Reality, as demonstrated with HoloLens, seamlessly integrates virtual and physical elements, enabling the robot to interact with both [37,72]. This breakthrough eliminates the need for real multi-sound sources, as virtual landmarks can be generated and placed in scenarios lacking physical landmarks. For detailed HoloSLAM implementation, please consult the reference [62].

In this project, the Microsoft HoloLens-Mixed Reality landmark-based SLAM (HoloSLAM) is utilized along with the ellipsoidal set-membership filter method [11] to address the challenges associated with landmarks in landmark-based acoustic-based SLAM. This approach allows for accurate robot localization and mapping even without multiple sound sources are required. With HoloSLAM, the robot gains the capability to dynamically place virtual landmarks within its environment in real time, using its robot voice as a means of interaction. A virtual landmark represents a digital entity serving as a recognizable point or feature in augmented or mixed-reality environments. It includes items like 3D models, holographic representations, or interactive elements strategically placed in the robot’s physical space, seamlessly blending with reality to enrich its environment. These virtual landmarks play a crucial role in the robot’s accurate self-localization, eliminating the need for explicit sound source locations.

Devices with powerful CPUs and GPUs are required for processing these virtual landmarks (the virtual digital data) and real-world information, alongside display devices like lenses or screens to showcase generated digital content, facilitating immersive environments. Common extended reality devices include Microsoft HoloLens, Magic Leap One, Epson Moverio, and Google Glass, while popular VR choices encompass HTC Vive, Oculus Quest, Valve Index, and Sony PlayStation VR. Additionally, companies like Microsoft offer HMD display devices for mixed-reality production, alongside various smart glasses [62].

The HoloSLAM with Mixed Reality and Microsoft HoloLens operates within a framework defined by two distinct scenarios. In the first scenario, real landmarks are fully accessible and detectable, providing the robot with tangible points of reference for navigation. In contrast, the second scenario arises when real landmarks are either unavailable or undetectable, necessitating the reliance on virtual landmarks presented through the device to facilitate navigation. The audio-based HoloSLAM system is specifically crafted to function solely with virtual landmarks.

HoloLens seamlessly integrates virtual elements into the real world utilizing spatial mapping and tracking technology. This innovative process incorporates advanced sensors, cameras, and algorithms to ensure that virtual objects maintain their position and perspective within the environment, adapting to changes in the user’s viewpoint or location. Through depth cameras and IMUs, HoloLens constructs a detailed 3D representation of the surroundings, continuously updating it to reflect any alterations. This spatial mapping capability distinguishes HoloLens as a mixed-reality device, setting it apart from standard augmented reality tools.

Spatial mapping involves generating a three-dimensional depiction of the physical space, while scene understanding interprets the elements within it, recognizing objects and their attributes. In Unity, this is achieved by creating a 3D mesh using depth data from the camera, representing surfaces in the environment. Each surface triangle is linked to a world-locked spatial coordinate system, ensuring consistency in virtual object placement [37].

Despite the complexity, HoloLens adjusts virtual objects in real-time based on robot head movement and perspective changes, applying perspective corrections to maintain realism. However, challenges such as variations in perspective and lighting, as well as occlusions, can affect visual integration. Microsoft has not officially disclosed details regarding algorithms and hardware precision, but practical experiments suggest an accuracy within a few centimeters. This margin of error is considered in implementations like audio-based HoloSLAM. HoloLens offers a compelling augmented reality experience through its sophisticated integration techniques.

Upon successfully deploying an audio-based virtual landmark, the robot proceeds to localize itself within the environment and seamlessly incorporates this landmark into the existing map, establishing it as a pivotal reference point for subsequent navigation tasks. These virtual landmarks, displayed through the HoloLens, serve as surrogate points of reference, enabling the robot to continue its navigation task despite the absence of real audio cues.

Figure 5 depicts the typical Unity3D engine interface showcasing the HoloSLAM virtual landmark hologram application.

The following is the pseudo-code for the hologram app that employs the HoloLens mixed reality technique to place the virtual landmarks in the robot environment. The execution of this hologram occurs exclusively within the HoloLens device and is triggered solely by a voice command (***Start***) generated through the robot’s speakers.
***Start-launch***  *the Holo-landmark hologram app.*
            *voice function command (Start)*
***Place Virtual Landmark Observation***–                 *-Voice function command (place virtual landmark).*
***Take a Picture (if needed)*** –   *-voice function command (takepicture).*
***Exit-***  *close the Holo-landmark hologram app.*
        *voice function (Exit)*


Once the Holo-landmark hologram app is deployed in the HoloLens device through the robot’s voice command function (***Start***), the robot initiates the HoloLens to ensure the app is operational. Subsequently, when the active speaker is recognized and the robot receives a movement command, it prompts the Holo-landmark hologram to place virtual landmarks in predetermined positions using the voice command (***place virtual landmark***) through the Holo-landmark hologram app. In this paper, the Holo-landmark hologram app is purpose-built to introduce a singular type of virtual landmark, such as a cube, sphere, or diamond shape, with each execution of the voice command “***place virtual landmark***”. Different applications require different functions and different virtual landmarks. Nonetheless, the Holo-landmark app can easily incorporate additional objects or landmarks by simply incorporating new functions. Unity3D, Morphi, 3D Slash, Fusion 360, and Blender are valuable resources that provide a vast collection of pre-existing 2D/3D objects that can serve as audio virtual landmarks [73]. These virtual landmarks can be placed in random positions, offering flexibility and freedom in selecting their desired locations. The functionalities embedded within the virtual holographic application on the HoloLens may vary across different applications, offering flexibility in the features available for navigation assistance. However, irrespective of the complexity or simplicity of the audio-based navigation task, the fundamental design principles governing the creation and utilization of virtual landmarks remain consistent. These virtual landmarks play a crucial role in facilitating audio-based virtual HoloSLAM, ensuring robust and reliable navigation capabilities under varying environmental conditions.

Presented below is the pseudo-code for the integration of an audio-based HoloSLAM hologram with Ellipsoidal-SLAM. The core structures of Ellipsoidal-SLAM remain intact, while the HoloLens-Mixed Reality operations come into play when a virtual landmark is placed. These virtual landmarks are then incorporated into the global location mapping using Ellipsoidal-SLAM.
***Start******Initialization****—SLAM Initialization, NAO Robot Initialization, **Launch Holo-landmark hologram app (******voice function command(start)).******Get Observation (4)****–**Is the Active Speaker Identified?******Yes****—**Holo-landmark** hologram app**-Place Virtual Landmark **(******voice function command (******place virtual landmark******))******No****—**No** action (wait for speaker identification).**while not_stop****Prediction Step****—Check for a safe distance to move by **sonar**. (****Move command****)****No****—safe distance. Turn 180 degrees.****Is the Active Speaker Identified?******Yes****—**Holo-landmark** hologram app**-Place Virtual Landmark (****voice function command (******place virtual landmark******)****)****No****—No action (wait for speaker identification)****Data Association(5)-******Virtual Landmark***  *matching and data-association simplification*
***Correction_Step—****Run standard Ellipsoidal—SLAM update step.****Augmented_Map******—****Add new **Virtual Landmarks** to the map**Check if iteration numbers are achieved.**No—Go to step 4****End****—**Close-Holo-landmark hologram app.*** ***Voice function command (stop)***

During the initialization process of Ellipsoidal-SLAM, the robot simultaneously initiates the Holo-landmark hologram app by issuing a voice command (start). If the robot detects an active speaker through the detected keyword commands, it responds to their command by placing a virtual landmark through the voice function command (place virtual landmark) first and then responds to the movement command.

The primary contribution of integrating HoloLens-Mixed Reality into sound-based Ellipsoidal-SLAM lies in the utilization of virtual landmarks in situations where multiple sound sources or acoustic landmarks are unavailable, and the full SLAM process cannot be accomplished.

### 3.4. Active Speaker Representation and Modeling

Human speech conveys various information like words, emotions, gender, and identity. Initially, speech comprehension outpaced speaker identification [74]. Speaker recognition research started in the 1960s, notably by Pruzansky and Mathews in 1964, who used digital spectrograms to verify speakers [75]. This formed the foundation for extensive research in speaker recognition, investigating various feature extraction and similarity measurement techniques. Automatic speaker recognition, or voice biometrics, identifies individuals using vocal traits. It has applications in security, forensics, and human–computer interaction. Unlike speech recognition, it focuses on verifying or identifying speakers, involving **speaker verification** to confirm claimed identity and **speaker identification** to determine identity from a group of candidates [76,77].

Speaker recognition systems have core components: **feature extraction** and **feature matching**. Extraction involves capturing data from the voice signal to represent speakers. Matching identifies unknown speakers by comparing their features to known speakers. This field has text-dependent and text-independent recognition. Text-dependent recognition requires accurate utterance of a password, while text-independent recognition verifies identity without content restrictions [42].

This study targets active speaker identification, employing MFCC and GFCC. Multiple recognition methods, including HMM, GMM, RF, SOM, statistical approaches, and deep neural networks, are applicable. GMM, SVM, and deep neural network-based classifiers were tested for active speaker classification.

#### 3.4.1. Feature Extraction Techniques for Active Speaker Identification

Audio feature extraction, part of signal modeling, transforms audio data into a domain that groups similar instances and separates distinct categories. Inspired by human auditory and articulatory systems, these methods yield meaningful representations. Effective feature extraction reduces data dimensionality, offering several benefits such as decreased computational complexity and the removal of redundant or irrelevant information [78]. This study utilizes GFCC and MFCC features for speaker identification, as they capture diverse voice signal characteristics. Figure 5 visually depicts the MFCC and GFCC feature extraction processes. Both share a common sequence of stages, differing only in filter bank types applied to the frequency domain signal obtained through FFT and the subsequent compression step. MFCC uses a Mel filter bank, followed by logarithmic compression and DCT. In contrast, MFCC uses a Mel filter bank, followed by logarithmic compression and Discrete Cosine Transform (DCT). In contrast, GFCC applies a Gammatone filter bank, pre-loudness, and DCT. Figure 6. shows the Block diagram of MFCCs and GFCCs feature extraction modules.

The MFCC feature vectors target human speech frequencies up to 1000 Hz using linear and logarithmic filters, capturing spoken word spectral characteristics precisely.

For the MFCC feature, let us consider *X*(*n*) as the original input speech signal, and *Y*(*n*) as the enhanced or amplified speech signal, given by Equation (14):(14)$$Y\left(n\right)=X\left(n\right)-a*X(n-1)$$

The pre-emphasis factor, typically chosen from the range of 0.95 to 0.98, is applied. Subsequently, a smoothing window function, such as Hamming windows (Equation (15)), is used on the pre-emphasized speech signal *Y*(*n*).
(15)$$W\left(n\right)=0.54-0.46*cos\;cos\left(\frac{2\pi n}{N-1}\right),\;0\leq n<N-1$$

The time-domain signal is then transformed to the frequency domain using Fast Fourier Transform (FFT). A Mel filter bank, designed for speaker recognition, refines the spectrum. In the final step, the log Mel spectrum is converted back to time, yielding MFCC using logarithmic compression and discrete cosine transform (Equation (16)).
(16)$$C_{n}=\sum_{m}^{M}\left[log\;log\;S(m)\right]\;cos\;cos\left[\frac{\pi n}{M}\left(m-\frac{1}{2}\right)\right],\;0\leq n<N-1$$

M represents the output of an M-channel filter bank, and ***n*** is the index of the cepstral coefficient. This cepstral representation effectively captures the local spectral properties of the signal for the given frame analysis.

To analyze transitions, another method involves calculating the first difference of MFCC signal features, referred to as the feature’s delta (Δ*f*). This delta signifies the rate of change in a feature over time, providing insights into transitions between speech sounds. It is computed using a simple formula:$${\Delta f}_{k}=(f_{k+1}-f_{k-1})/2$$
where $f_{k+1}$ and $f_{k-1}$ are feature values at adjacent time points. Figure 6 displays the MFCC, Delta, and Delta-Delta features in our dataset. These features provide hierarchical representations of audio signals, capturing spectral characteristics, temporal dynamics, and higher-order temporal variations, respectively. The output graph illustrates that MFCC-delta-delta contains fewer coefficients compared to both MFCC-delta and MFCC. The figure generates a more comprehensive depiction of the frame’s context, resulting in improved accuracy. In this representation, the x-axis denotes time, while the y-axis represents the MFCC coefficient values.

GFCC’s feature computation parallels MFCC’s with a significant distinction: the use of gammatone filters. These filters, inspired by human cochlear processing, enhance feature extraction from the FFT spectrum. Like MFCCs, the process involves pre-emphasis, windowing, and FFT. Gammatone filters are then applied, extracting distinct features for GFCC representation, and capturing auditory system-inspired insights. The formula representing each filter’s impulse response is as follows [39]:(17)$$g\left(t\right)=at^{n-1}e^{-2\pi bt}\;cos\;cos\;(2\pi f_{c}t+\varphi )$$

As ‘*a*’ remains constant, ‘*n*’ and ‘***φ***’ are consistent throughout the filter bank. The Gammatone filter bank’s frequency selectivity relies mainly on two parameters: central frequency ‘*f*’ and filter bandwidth ‘*b*’. A common approach for setting these values approximates them based on human cochlear filters using the Equivalent Rectangular Bandwidth (ERB), following Moore’s model [79]. This approach effectively simulates the human auditory system.
(18)$$ERB\left(f_{c}\right)=24.7+0.108\;f_{c}$$

To align the Gammatone filter with human auditory characteristics, follow Moore’s recommendation [79] and Patterson et al.’s use of the ERB concept [80]. Set the filter parameters as bandwidth (*b*) = **1.019 ∗ ERB** and filter order (*n*) = 4. This ensures better compatibility with the human auditory system. Moore’s guidance suggests spacing center frequencies uniformly on the ERB frequency scale, creating the relationship between the number of ERBs and corresponding center frequencies, denoted as $f_{c}$, which can be represented by the following expression:(19)$$numberofERBs=21.4\;{log}_{10}(0.00437\;f_{c}+1)$$

The ERB scale, logarithmic in nature, relates center frequencies and frequency energy distribution in speech, following a $\frac{1}{f}$ pattern. Gammatone filters adapt bandwidth, narrowing at lower frequencies and broadening at higher frequencies. Figure 7 displays the MFCC, Delta, and Delta-Delta features in our dataset. Figure 8 illustrates the cochleagram response, a Gammatone filter bank, and a typical spectrogram in response to our dataset.

#### 3.4.2. Active Speaker Representations (Classification Algorithms)

Following the extraction of distinct features from an audio signal, a classifier is utilized to differentiate these features, creating a model for each speaker. These models are then used to compare new input features with stored reference templates to determine identity. Speaker classification involves **stochastic** (parametric) models like Gaussian mixture models and Hidden Markov Models, which use probabilistic pattern matching, and **template** (non-parametric) models like Dynamic Time Warping and Vector Quantization, which employ deterministic pattern matching [81,82]. The choice of classification method depends on the specific application, with dynamic time warping and hidden Markov models suited for text-dependent recognition and vector quantization and Gaussian mixture models commonly used for text-independent recognition. This section covers established classification algorithms extensively used in speech recognition and active speaker identification in this work.

##### Gaussian Mixture Model

In this section, we clarify the structure of the GMM and its rationale for representing active speaker identity in text-independent speaker identification. GMM is a robust tool in Speaker Recognition Evaluations (SREs), adept at addressing data analysis and clustering challenges through a mixture of Gaussian densities. Through unsupervised techniques like clustering, GMM offers a valuable probabilistic model for data grouping. In contemporary text-independent GMM systems, the Expectation Maximization (EM) algorithm is commonly used to estimate background model parameters, ensuring GMM-based methods remain at the forefront of speaker recognition advancements. The GMM model is defined as a likelihood function with a mixture of *M* Gaussians, expressed by the following equation [40]:(20)$$p\left(\vec{x}|\lambda \right)=\sum_{i=1}^{M}w_{i}p_{i}(\vec{x})$$
where $p\left(\lambda \right)$ is the frame-based likelihood function, λ is the hypothesis or likelihood function, x is a set of features (MFCCs or GFCCs), and (***x***) is the individual Gaussian density function.

The model is estimated by a weighted linear combination of D-variate Gaussian density function $p_{i}(\vec{x})$, each parameterized by a mean D×1 vector, $\mu_{i}$, mixing weights, which are constrained by $w_{i}\geq 0$, $\sum_{i=1}^{M}w_{i}=1$, and a D×D covariance matrix, $\Sigma_{i}$ as:(21)$$p_{i}(\vec{x})=\frac{1}{{2\pi }^{D/2}{\left|\Sigma_{i}\right|}^{1/2}}exp\{\frac{1}{2}{\left(x-\mu_{i}\right)}^{\prime }{{(\Sigma }_{i}}^{-1})(x-\mu_{i})\}$$

Once the model has completed its training, the subsequent step involves evaluating the log-likelihood of this model using a test set comprising feature vectors MFCCs/GFCCs.
(22)$$p\left(\lambda \right)=\sum_{i=1}^{M}p\left(\lambda \right)$$

##### Support Vector Machine (SVM)

SVM, a binary classifier, distinguishes speakers from impostors via a separation hyperplane. Exploring SVM techniques assesses novel classification methods for speaker identification, enhances comprehension of the challenge, and determines whether SVMs offer insights beyond traditional GMM approaches. SVM utilizes a kernel function to create a binary classifier, with the sequence kernel based on generalized linear discriminants. Notably, it directly expands into the SVM feature space while maintaining computational efficiency and increased accuracy. SVM complements and competes effectively with other methods, including Gaussian mixture models. It seeks the optimal hyperplane that maximizes the margin between data and the separation boundary, resulting in the best generalization performance [41]. Figure 9 shows the principle of the optimal hyperplane and the optimal margin in SVM modeling.

The discriminant function of the SVM is given by:(23)$$ƒ\left(x\right)=\sum_{i=1}^{N}\alpha_{i}t_{i}K\left(x,x_{i}\right)+d$$
where the $t_{i}$ are the ideal outputs, $\sum_{i=1}^{N}\alpha_{i}t_{i}=0$, and *α_i_* > 0. The vectors $x_{i}$ are support vectors and are obtained from the training set by an optimization process. The ideal outputs are either 1 or −1, based on the support vector class. The kernel K(. , . ) is constrained to have certain properties (the Mercer condition), so that K(. , . ) can be expressed as:(24)$$K\left(.,.\right)={b\left(x\right)}^{t}b\left(y\right)$$
$b\left(x\right)$ maps input space in SVM, where a two-class model for speaker identification is trained. The known non-targets comprise the second class, with class 0 assigned to the target speaker’s utterances.

SVM can be represented as a two-class problem: target and nontarget speaker. If ω is a random variable representing the hypothesis, then ω = 1 represents the target being present and ω = 0 represents the target not being present. A score is calculated from a sequence of observations $y_{1},\ldots ,y_{n}$ extracted from the speech input. The scoring function is based on the output of a generalized linear discriminant function of the form $g\left(y\right)=\omega^{t}b\left(y\right)$ where ω is the vector of classifier parameters and b is an expansion of the input space into a vector of scalar functions [33]:(25)$$b\left(y\right)={\left[b_{1}\left(y\right)b_{2}\left(y\right)\ldots .b_{n}(y)\right]}^{t}$$

If the classifier is trained with a mean-squared error training criterion and ideal outputs of 1 for ω=1 and 0 for ω=0, then $g\left(y\right)$ will approximate the posterior probability p(ω=1|y)**.** We can then find the probability of the entire sequence, $p(y_{1}\ldots y_{n}|\omega =1)$ as follows:(26)$$p\left(y_{1}\ldots y_{n}|\omega \right)=\prod_{i=1}^{n}p\left(y_{i}|\omega \right)=\prod_{i=1}^{n}\frac{p\left(\omega |y_{i}\right)p\left(y_{i}\right)}{p\left(\omega \right)}$$

Taking log on both sides [33], we obtain the discriminant function:(27)$$d^{\prime }\left(y_{1}^{n}|\omega \right)=\sum_{i=1}^{n}\log\left(\frac{p\left(\omega |y_{i}\right)}{p(\omega )}\right)$$

For classification purposes, we discard $p\left(y_{i}\right)$. Using log⁡(x) ≈x−1:(28)$$d\left(y_{1}^{n}|\omega \right)=\frac{1}{n}\sum_{i=1}^{n}\log\left(\frac{p\left(\omega |y_{i}\right)}{p(\omega )}\right)$$

Assuming $g\left(y\right)\approx p(\omega =1|y)$:(29)$$d\left(y_{1}^{n}|\omega =1\right)=\frac{1}{n}\sum_{i=1}^{n}\log\left(\frac{w^{t}b(y_{i})}{p(\omega =1)}\right)=\frac{1}{np(\omega =1)}w^{t}\left(\sum_{i=1}^{n}b(y_{i})\right)=\frac{1}{p\left(\omega =1\right)}w^{t}{\bar{b}}_{y}$$
where the mapping $y_{1}^{n}\to b_{y}$ by is:(30)$$y_{1}^{n}=\frac{1}{n}\sum_{i=1}^{n}b\left(y_{i}\right)$$

In the scoring method, for a sequence of input vectors $x_{1},x_{2},\ldots .,x_{n}$ and a speaker model w, we can construct ***b*** using (30). For speaker identification, if the score is above a threshold, then we declare the identity claim valid; otherwise, the claim is rejected as an impostor attempt.

##### Deep Learning-Based Models Architecture

In recent years, deep learning-based models have become the cornerstone for audio classification tasks, enabling the automatic categorization of audio signals into various classes, such as speech recognition, music genre classification, and environmental sound analysis. These models, characterized by their sophisticated architectural design, have demonstrated remarkable performance in handling complex audio data, making them an indispensable tool in various domains including multimedia analysis, content recommendation, and surveillance systems. DNN excels here by leveraging multiple filters during training to extract unique features from input spectrograms. These features improve the representation of active speakers in speech data, autonomously learned and then used for identification by a classifier [83]. In this section, we explore CNN-LSTM and TDNN architectures as the two main ones that have been employed in this work.

**1.** 
**Convolutional Long Short-Term Memory Network**


CNN, a deep learning model based on convolution, is primarily used for image analysis in machine learning. However, it has shown broad utility in recognizing audio patterns, improving images, processing natural language, and forecasting time series data. The CNN architecture, introduced by Lecun et al. [84], consists of an input layer, an output layer, and concealed layers, with convolutional layers performing dot product operations between input matrices and convolutional kernels.

A Long Short-Term Memory (LSTM) Network belongs to the category of recurrent neural networks (RNNs), which are essentially neural networks with feedback loops [85]. RNNs perform well in speech recognition, language modeling, and translation, but they face a key challenge: the vanishing gradient problem. This occurs when the error gradient dwindles or grows explosively during backpropagation, especially across multiple time steps, leading to limited memory capacity, often called ‘**short memory**’. LSTM network architecture offers a solution by using a special memory cell to control information flow. This selectively retains or discards data, preventing gradient problems and enabling the learning of long-term dependencies in sequential data. Figure 10 illustrates the architecture of an LSTM cell. Each cell receives two critical inputs: the output sequence produced by the previous LSTM cell and the hidden state value from the previous cell, denoted as $h_{t-1}$. Inside the cell, there are three gates: the forget gate $f_{t}$, the input gate $i_{t}$, and the output gate $O_{t}$.

Information from the previous hidden state $h_{t-1}$ and information from the current input $x_{t}$ are passed through the sigmoid function. The forget gate acts as a filter to forget certain information about the state of the cell. To this end, a term-to-term multiplication is carried out between $f_{t}$ and $c_{t-1}$, which tends to cancel the components of $c_{t-1}$ close to 0. A filtered cell state is then obtained as follows:(31)$$f_{t}=\sigma (W_{f}\left[h_{t-1},x_{t}\right]+b_{f}$$
where ***σ*** denotes the sigmoid activation function, which is a nonlinear function that maps its input to a value between 0 and 1$,\;W_{f}$ is the weight of the forget gate, and $b_{f}$ is the bias. The weights and bias values are acquired through the training process of the LSTM.

LSTM employs the input gate for data integration into the memory cell, comprising the input activation gate and the candidate memory cell gate. The input activation gate manages data integration, while the candidate memory cell gate governs data storage within the memory cell.

By considering both the previous hidden state $h_{t-1}$ and the current input node $x_{t}$, the input gate in an LSTM generates two essential vectors: the input vector $i_{t}$ and the candidate memory cell vector ${\tilde{c}}_{t}$. Equation (32) describes the operation of the input activation gate, which involves the weight matrix $W_{i}$ and bias vector $b_{i}$. Simultaneously, Equation (33) demonstrates the formation of the candidate memory cell ${\tilde{c}}_{t}$ by applying the hyperbolic tangent activation function (tanh) to the same set of inputs, utilizing the weight matrix $W_{c}$ and bias vector $b_{c}.$
(32)$$i_{t}=\sigma \left(W_{i}\left[h_{t-1},x_{t}\right]+b_{i}\right)$$
(33)$${\tilde{c}}_{t}=tanh(W_{c}\left[h_{t-1},x_{t}\right]+b_{c})$$

The input vector and the candidate memory cell vector are merged to update the previous memory cell $c_{t-1}$, as shown in Equation (34). In this equation, the symbol **⊙** represents element-wise multiplication.
(34)$$c_{t}=f_{t}⊙c_{t-1}+i_{t}⊙{\tilde{c}}_{t}$$

The output gate controls data transfer from the memory cell to the current hidden state, serving as the LSTM’s output. The output gate vector $o_{t}$ is computed with this equation:(35)$$o_{t}=\sigma \left(W_{o}\left[h_{t-1},x_{t},c_{t}\right]+b_{o}\right)$$

Subsequently, the current hidden state, $h_{t}$, is derived using following equation:(36)$$h_{t}=o_{t}⊙tanh\left(c_{t}\right)$$

The new cell state and that of the hidden state are then directed to the next time step. Training sequential neural networks minimizes loss over data sequences using backpropagation through time (BPTT) for temporal gradients. Weight updates are computed mathematically based on loss function L and the learning rate η can be expressed as:(37)$$\Delta W=-\eta \frac{\partial L}{\partial W}$$

In this paper, the CNN-LSTM architecture utilizes CNN layers to construct a model of an active speaker from input data to enhance the model’s ability to make sequence predictions.

**2.** 
**Time-delay neural networks (TDNNs)**


A Time-Delay Neural Network (TDNN) is a dynamic network designed to capture temporal relationships between events and maintain temporal translation invariance. Initially introduced to enhance modeling of extensive temporal context [43], TDNN models have found applications in spoken word and online handwriting recognition. TDNN remains a common choice for acoustic modeling in modern speech recognition software such as Kaldi [84]. Its primary function is to convert acoustic speech signals into sequences of phonetic units, known as ‘phones’. The network takes acoustic feature frames as input and produces output depicting probability distributions for each phonetic unit. The network takes acoustic feature frames as input and produces a probability distribution for a defined set of target language phones. The goal is to classify each frame into the phonetic unit with the highest likelihood. In a single TDNN layer, each input frame is represented as a column vector, symbolizing a time step, with rows representing feature values. A compact weight matrix, often called a kernel or filter, slides over the input signal, performing convolution to generate the output.

Consider an input vector $x_{t}$ in $R^{m}$ as a matrix containing m numerical values, such as amplitudes at a specific frequency or the values of acoustic features within a filter bank bin. At each time step t, we will have a matrix of input features ***X*** ∈$R^{m\;x\;t}$ where each vector represents one-time step t of our speech signal with a trainable weight matrix ***W*** ∈$R^{m\;x\;l}$, where the kernel maintains a consistent height of m and a width of l, as illustrated in Figure 11.

The kernel W moves across the input signal with a space of s making s steps in each movement. The area on the input feature map that the kernel encompasses is termed the “receptive field.” Depending on the specific implementation, it is possible that the input may be filled with null values at both ends of height of m and a length of p. The output width o, resulting from the number of times the kernel can fit over the length of the input sequence, can be calculated as follows:(38)$$o=\left\lfloor \frac{t-l+2p}{s}\right\rfloor +1$$
where $\left\lfloor .\right\rfloor$ represents the floor function.

During each time step t, the TDNN conducts a convolution operation, which involves performing an element-wise multiplication (commonly known as the Hadamard product) between the kernel weights and the input located below it, followed by the summation of these resulting products.

Within the neural network, a trainable bias term b is included (which is not shown in the images above). The outcome is then processed through a non-linear activation function denoted as ϕ (examples of which include sigmoid, rectified linear, or p-norm functions). This process results in the formation of an output $z_{q}\;\epsilon \;z$ where z represents the entire output vector, achieved by performing this operation across all time steps (depicted as the light green vector in the images). Hence, the concise representation of the scalar output for a single element, denoted as $z_{q}\;\epsilon \;z$, at the q-th output step within the set {1,2…, o}, can be expressed as:(39)$$z_{q}=\phi (W*X_{q}+b)$$
where ∗ denotes the convolution operation and $X_{q}$ are the inputs in the receptive field. It can also be equivalently given by:(40)$$z_{q}=\phi (\sum_{i=1}^{m}\sum_{k=1}^{l}w_{i,k}x_{i,k}+b)$$

In this equation, the initial summation extends across the height of the acoustic features, while the subsequent summation covers the width of the receptive field or the width of the kernel. It is important to note that the kernel weights $w_{i,k}$ are shared across all output steps ***q***. Because the weights of the kernel are shared across the convolutions, the TDNN acquires a representation of the input that remains insensitive to the precise location of a phone within the broader sequence. Additionally, this sharing of weights reduces the quantity of parameters that need to be trained.

Considering that we need to repeat the same convolution operation as before, denoted as $z_{q}=\phi (W*X_{q}+b)$, it is important to note that the input vectors $X_{q}$ have also grown due to the expanded receptive field. In simpler terms, this process involves extracting the receptive field from its input, combining it, and applying the identical convolution operation.

Finally, in the context of employing multiple kernels, represented as H kernels, where each kernel is can be represented as $W^{\left(h\right)}\;\epsilon \;W^{\left(1\right)},\ldots ,W^{\left(H\right)}$, where each kernel similarly moves across the input. This process results in the generation of a sequence of output vectors, which can be structured into an output matrix **Z** $\epsilon \;R^{hxo}$.

Within a deep neural network architecture, this output can subsequently serve as the initial hidden layer of the network and be employed as input for the subsequent layer of the TDNN.

## 4. Experimental Results

### 4.1. Active Speaker Identification

In this section, we present the results of our implemented speaker identification system experiments. We conducted a performance comparison of five different classification methods: GMM, SVM, CNN, DNN, and TDNN. The evaluation is based on a dataset featuring two distinct active speaker classes, Speaker-1 and Speaker-2, recorded in a noise-free environment at the AISL. We used two feature extraction methods: one with MFCC and additional features, and the other with GFCC and similar features. Our primary aim is to assess the accuracy of each method in precisely distinguishing between these two speakers. The experimental analysis provides valuable insights into the strengths and weaknesses of each approach in the context of active speaker identification tasks.

The GMM was fitted using EM and clusters were determined using K-Means clusters. EM showed stable convergence. We explored different covariance types in GMM. Figure 12 and Figure 13 visualize clustering. GMM-MFCC features effectively grouped data. GMM-GFCC captured some speaker patterns but with lower accuracy. Future work can enhance feature extraction, use advanced clustering, or increase training data for better accuracy.

Table 1 compares how different covariance types in GMMs affect K-fold split data, showing their influence on GMM’s clustering ability. GMM clustering with GFCC features yielded 31% accuracy, while MFCC features improved accuracy to 80.9%. Although not ideal, the scatter plot suggests the model identified speaker similarities, offering the potential for further clustering enhancements.

Next, we reveal the outcomes of two SVM models employing MFCC and GFCC feature extraction. Our aim is to assess the effect of feature extraction on SVM performance. Both models used a linear kernel with C = 1.0. The SVM-MFCC model achieved 100% accuracy, demonstrating the power of these features for classification. In contrast, the SVM-GFCC model, while slightly less accurate, displayed robust classification abilities with some overlap between classes, indicating potential for refinement. These results highlight how feature extraction impacts SVM performance: MFCCs excel, while GFCCs offer solid performance with opportunities for improvement in separating classes. Figure 14 shows the decision boundary for the SVM model using MFCC features, clearly separating the two classes, confirming its 100% accuracy. In Figure 15 the decision boundary for the SVM model with GFCC features is more intricate, with some class overlap, explaining the slightly lower accuracy. However, it effectively separates most data points while acknowledging some overlap.

Visualizing decision boundaries provides insights into model behavior. The distinct boundary in the MFCC model highlights their suitability for this dataset. The GFCC model’s boundary, while effective, suggests potential benefits from additional feature engineering or model refinement for improved performance. Both SVM models with MFCC and GFCC show promise in dataset classification, with the choice depending on data characteristics and the balance between accuracy and interpretability.

We will now present the results of our speaker identification system using deep learning architectures, including Conv-LSTM, DNN, and TDNN models. These models were trained using combined features based on MFCC and GFCC, along with additional features such as chroma, Mel frequency, zero-crossing rate (ZCR), root mean square energy (RMSE), delta, and delta-delta features. Our goal is to evaluate how well these models accurately identify and distinguish between two active speakers. The recognition rate graph for the CNN, DNN, and TDNN models using MFCC features visually represents their performance in speaker identification. The graph displays changing recognition rates along the x-axis, reflecting their accuracy in identifying speakers. Early stopping was used to prevent overfitting.

Figure 16 displays the connection between training epochs and accuracy/recognition rate. It shows an initial steady increase in accuracy, signifying effective learning. However, there is a plateau, indicating diminishing returns with more training. Early stopping prevented overtraining by restoring the best weights, as evident in the stabilized accuracy curve.

Figure 17 reinforces accuracy findings by showing how the loss function changes over epochs. The loss aligns with accuracy, decreasing rapidly initially. However, like accuracy, it gradually levels off after a certain number of epochs, highlighting the ideal training duration. Early stopping effectively curbed loss deviation caused by overfitting.

The performance of the six models was graphically presented for easy comparison. Remarkably, all six models consistently reached a final training accuracy of 100%, as seen in Figure 18 (Recognition Rate) and Figure 19 (Loss Function Value).

The models’ high accuracy demonstrates their effectiveness in recognizing unique speaker traits from input features. Table 2 summarizes our thesis results. TDNN excelled in speaker classification compared to Conv-LSTM CNNs and traditional DNNs. Its specialized temporal sequence modeling effectively captured complex speech patterns, enhancing speaker discrimination accuracy.

The use of early stopping played a key role in achieving these results by ending training at 20 epochs, preventing overfitting by monitoring performance on a validation set. The models’ ability to stop early highlights their fast learning and efficient parameter optimization. The thorough evaluation of the six models (Conv-LSTM, TDNN, and DNN) using both MFCC and GFCC features yielded impressive outcomes. Perfect accuracies within a limited number of epochs showcase their ability to capture intricate speaker traits. Different convergence speeds emphasize each architecture’s efficiency with specific features. Early stopping safeguards against overfitting, enhancing model reliability. This highlights deep learning models’ potential for speaker classification while stressing the importance of model selection based on feature characteristics. Future research could explore interpretability and robustness in real-world scenarios.

### 4.2. Audio Ellipsoidal-HoloSLAM Algorithm

This study utilizes the Seeed ReSpeaker Core v2.0 microphone, tailored for voice interface tasks, with a 16 kHz sampling rate. Operating on the GNU/Linux system via the Arduino device, the ReSpeaker microphone array connects to the Nao robot’s USB port [86]. Integrated into the Nao robot head alongside Microsoft’s HoloLens, the microphone array is depicted in Figure 20.

Table 3 displays some parameters of the Nao robot utilized in these experiments. For further information regarding this robot, please refer to [38].

When an individual speaks within the vicinity of the microphone array, the microphones capture the sound, which is subsequently transmitted to the onboard ADC on the ReSpeaker. The resulting data are then processed on the Raspberry Pi board. Numerous experiments have been undertaken with various speakers positioned at different distances to calibrate the ReSpeaker microphone array and determine the ideal angle for effective communication with the robot. This initial investigation provided early observations on humans’ ability to gauge the direction of a voice, even in situations with clear speech angles, such as 0°. Evidently, someone positioned directly facing the ReSpeaker (i.e., 0° off-axis) would be regarded as the optimal orientation. The ReSpeaker is capable of capturing sound within a 5-m range. The variations in DOA estimation of the microphone pairs are consolidated to a median value, yielding a unified live DOA output. This illustrates that our microphone array system is capable of precisely determining the sound source location. The ReSpeaker microphone array system can pinpoint the sound source with an average deviation of 5 degrees. Such precise localization enables robots to effectively perceive their surroundings and make informed navigational decisions.

In audio-based navigation systems, the utilization of multiple microphones strategically placed throughout the indoor environment is paramount for accurate localization and mapping by the robot. With prior knowledge of the microphones’ locations, the robot can triangulate sound sources more effectively, thereby improving the accuracy of its localization capabilities.

However, the task of localizing the robot and mapping its environment becomes significantly more challenging and impossible when fewer microphones are employed, especially in the absence of prior knowledge regarding their locations. This can lead to inaccuracies in localization and mapping, potentially resulting in navigation errors and reduced overall performance.

The Microsoft HoloLens, with its Mixed Reality-based Ellipsoidal SLAM (**HoloSLAM**), facilitates the placement of virtual landmarks if needed within the robot’s environment via the Holo-landmark hologram app when necessary. The Virtual Landmark App offers a variety of three distinct virtual landmark types to choose from, including diamonds, spheres, and cubes.

This innovative technology enables the robot to interact with these landmarks dynamically, offering functionalities such as moving up, moving down, and moving right. Moreover, the robot possesses the capability to modify these virtual landmarks in real-time through voice commands. This feature empowers the robot to adapt and customize its audio environment according to changing requirements or unforeseen circumstances swiftly. This ultimately enhances its navigational capabilities and overall functionality.

The experiments detailed in this paper were conducted utilizing the Nao robot inside an SFU campus environment. The main goal of the experiment is to assess the effectiveness of audio-based virtual HoloSLAM in estimating the robot’s position and mapping its environment. This evaluation specifically focuses on the robot’s performance when equipped with a single microphone array designed to track the active speaker. The target or active individual traverses a realistic path scenario, with the robot tracking and following their voices.

In this experiment, the robot first identifies the active speaker and their direction, then turns toward them. Subsequently, it places random virtual landmarks in space, capturing images and removing them from their surroundings, as depicted in Figure 21. These virtual objects are consistently positioned 2 m toward the speaker. The virtual landmarks are then utilized to complete the SLAM process.

The robot is assigned the task of identifying the active speaker through a combination of pre-trained models and subsequently mapping its environment and localizing itself within it autonomously, without prior knowledge or human intervention. The robot is programmed not to follow the speaker unless a recognition rate of 93.75% is achieved by six models (GMM, SVM, Conv-LSTM, DNN, and TDNN) utilizing both MFCC and GFCC features, ensuring accurate identification of the correct active speaker.

After identifying the active speaker using a hybrid pre-trained model, the robot utilizes ReSpeaker’s sound source localization to estimate the speaker’s direction. This angle guides the robot in orienting itself towards the speaker. Through a virtual hologram app, the robot instructs the HoloLens to place random virtual landmarks within its environment, depicted in Figure 21. These landmarks are consistently positioned 2 m along the x-axis of the Nao robot. This capability enables the robot to precisely position and remove virtual landmarks as needed, facilitating real-time communication with them. Moreover, it grants the robot greater autonomy and control over its mapping process. Consequently, even if the robot’s sensors fail to detect any landmarks during observation, the audio-based SLAM algorithm remains reliable.

Figure 22 and Figure 23 provide visual representations of the estimated robot position and virtual landmarks obtained using the audio-based virtual Ellipsoidal-HoloSLAM system.

Upon comparing the estimated positions with the actual robot position, it becomes evident that the newly implemented SLAM technology has adeptly tracked the robot’s movement and accurately constructed virtual landmarks with minimal margin for error.

Table 4 offers an intricate breakdown of outcomes garnered from real-time experiments aimed at assessing the efficiency of the implemented system. These findings offer valuable insights into the overarching performance of the audio-driven virtual Ellipsoidal HoloSLAM algorithm, particularly regarding its accuracy and reliability. Additionally, the evaluation extends to scrutinizing the estimated positions of the virtual landmarks.

The data within the table highlight a notable observation: the IMU of the Nao exhibits the most significant error concerning both the robot’s position and orientation. However, through the utilization of Ellipsoidal HoloSLAM, these errors are mitigated, showcasing a substantial improvement in localization accuracy. Notably, the algorithm consistently succeeds in accurately localizing and tracking the active speaker and building a map of the unknown environment at each iteration, indicative of its robustness and efficacy in real-time scenarios.

The new audio-based SLAM enables precise estimation of the robot’s position and meticulous mapping of its virtual environment at each stage of operation. However, a notable bottleneck emerged during active speaker identification, prolonging the process and highlighting the imperative for optimization to augment the model’s operational speed and efficiency.

## 5. Discussion

This paper advances the current state-of-the-art in audio-based SLAM by introducing a novel integration of a microphone array platform with Microsoft HoloLens, a robotic mixed reality device. The approach eliminates the need to prepare the robot environment with multiple audio sources and audio landmarks to perform complete and successful audio-based SLAM in indoor environments. This approach can operate with a single audio source and a solitary microphone array, ensuring precise localization of both the audio sources and the robot. The study utilizes a pre-trained or voice-printed speaker as the target audio source for the robot to follow and interact with. It also facilitates the mapping of audio landmarks to the robot’s environment, addressing the challenges associated with multiple audio sources and landmarks in indoor settings. Additionally, this approach successfully maps the environment with virtual landmarks, providing a comprehensive solution to the complexities associated with audio-based SLAM in indoor settings.

## 6. Conclusions

In this study, we proposed an audio-based SLAM system integrated with the Microsoft HoloLens mixed reality device to enhance the capabilities of intelligent robots. The main objective of this system is to conduct audio-based SLAM with minimal auditory requirements, presenting a novel perspective through the HoloLens and robotic mixed reality concept. The proposed system operates in several stages. Firstly, it leverages the audio features to identify a unique speaker by employing pre-registered voiceprints through deep learning in a multi-audio environment, utilizing a microphone array. The extracted audio is then utilized to estimate the direction of the speaker. Subsequently, the robot utilizes this estimated direction to track the active speaker while simultaneously localizing itself and generating a map of its surroundings. Due to the limited availability of audio landmarks, the Ellipsoidal HoloSLAM incorporates virtual landmarks into the mapping process. This inclusion allows for an accurate and realistic SLAM implementation without the need for prior knowledge of sound source locations. As the robot moves and the location and direction of the active speaker change, the implemented audio HoloSLAM algorithm continuously updates the robot’s position and orientation within the built map. This enables the robot to dynamically follow the speaker and simultaneously construct a detailed virtual map of the environment.

A comparative analysis with state-of-the-art audio-based SLAM systems revealed that the audio HoloSLAM achieved more accurate trajectories for the robot without the addition of extra sensors or reliance on additional audio landmarks or pre-known locations of audio sources. Real-world experiments were conducted to validate the implemented audio HoloSLAM system. The results demonstrated that the audio-based virtual HoloSLAM algorithm successfully mapped the environment and exhibited a more robust robot trajectory. The system accurately estimated the robot’s position at each movement with minimal errors. This approach exhibits significant potential in various indoor applications, including human–robot interaction, assistive robotics, and indoor navigation. The successful integration of the Microsoft HoloLens mixed reality device with audio-based SLAM opens up new possibilities for enhancing the spatial awareness and interaction capabilities of intelligent robots in various environments.

## Figures and Tables

**Figure 1 sensors-24-02796-f001:**
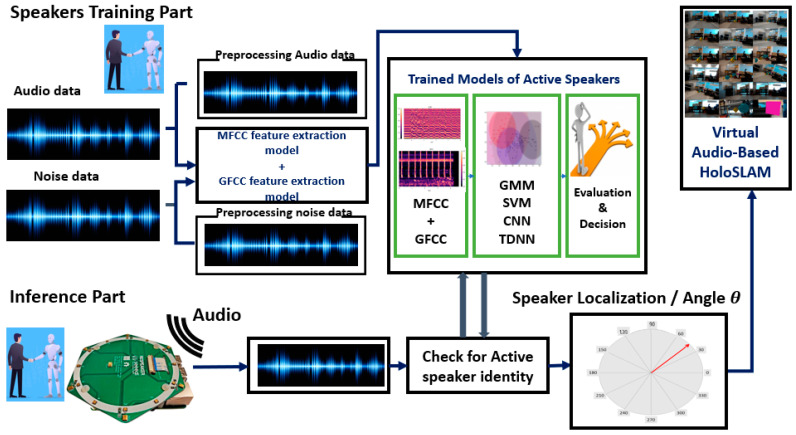
System overview of real-time virtual HoloSLAM process and active speaker identification and localization.

**Figure 2 sensors-24-02796-f002:**
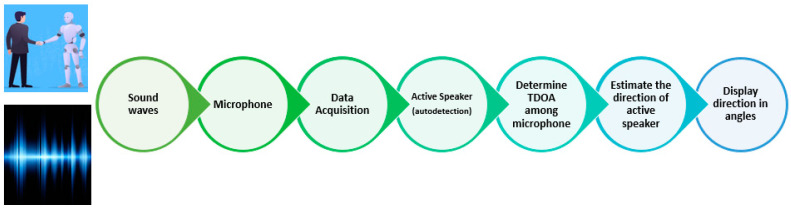
SSL design steps.

**Figure 3 sensors-24-02796-f003:**
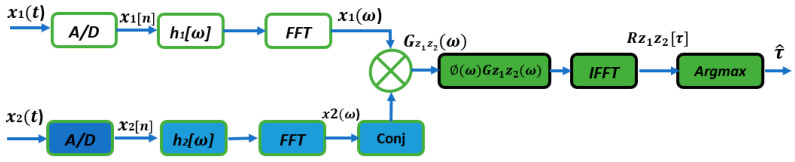
The Generalized Cross-Correlation-PHAT block diagram.

**Figure 4 sensors-24-02796-f004:**
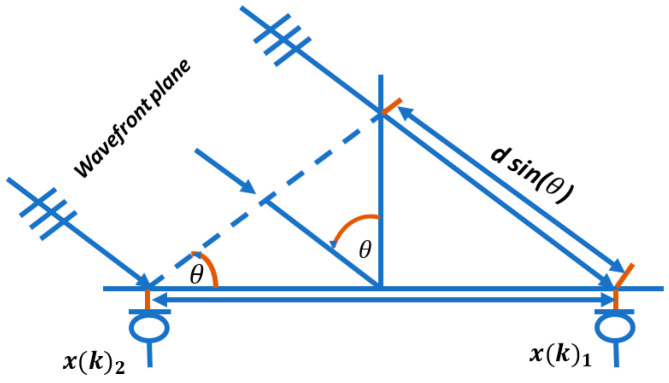
Calculating the angle of arrival between two microphones via ***τ***.

**Figure 5 sensors-24-02796-f005:**
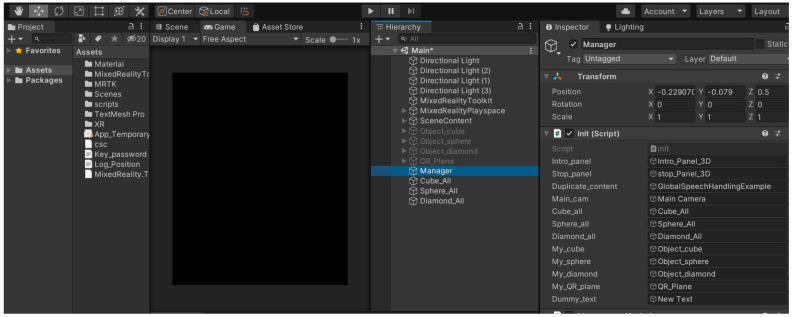
The Unity3D window of the HoloSLAM virtual landmark hologram app [62].

**Figure 6 sensors-24-02796-f006:**
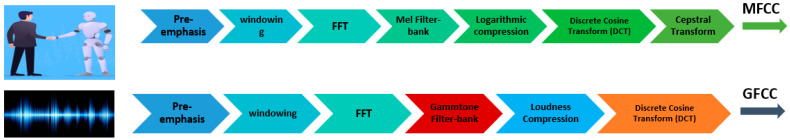
Block diagrams of MFCC and GFCC feature extraction modules.

**Figure 7 sensors-24-02796-f007:**
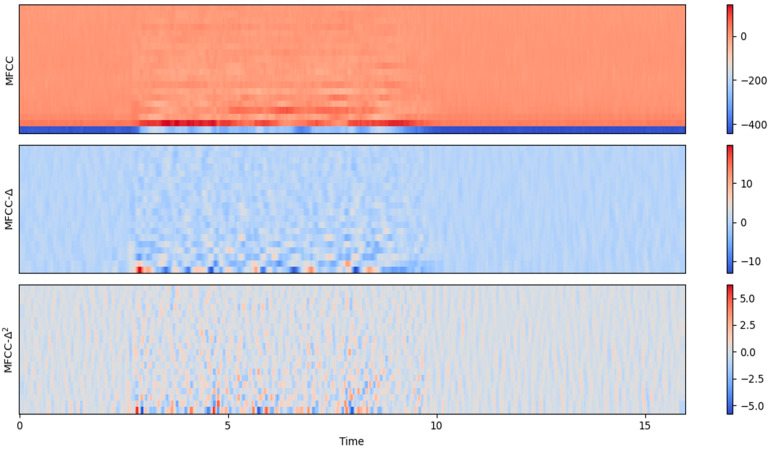
MFCC output of our dataset.

**Figure 8 sensors-24-02796-f008:**
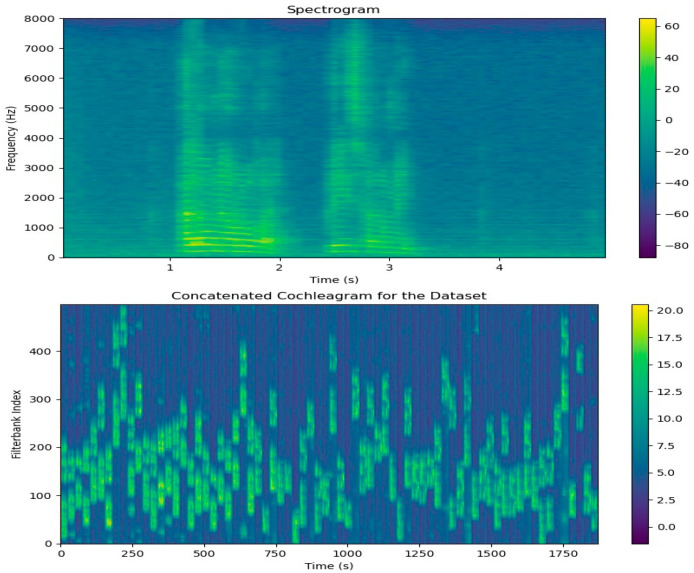
The Spectrogram and Cochleagram of a sample speech signal from our dataset.

**Figure 9 sensors-24-02796-f009:**
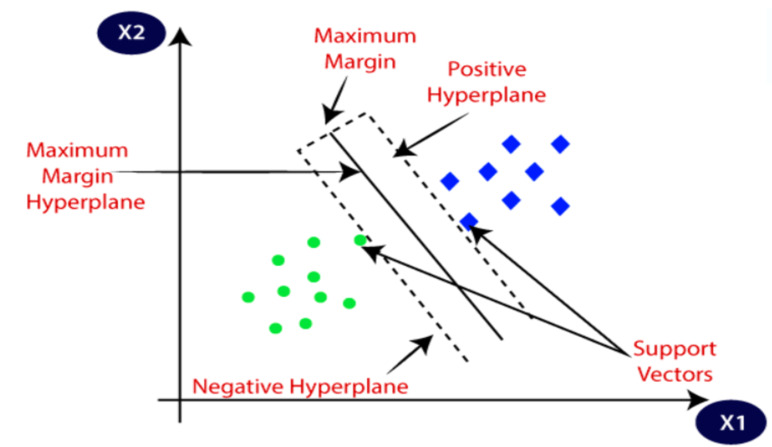
Principle of support vector machines.

**Figure 10 sensors-24-02796-f010:**
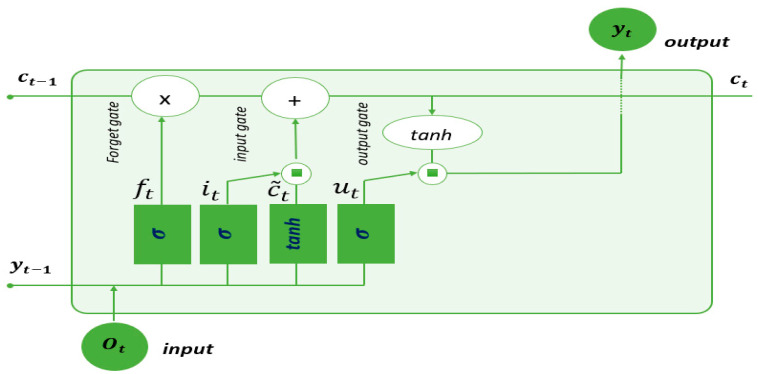
The architecture of an LSTM cell.

**Figure 11 sensors-24-02796-f011:**
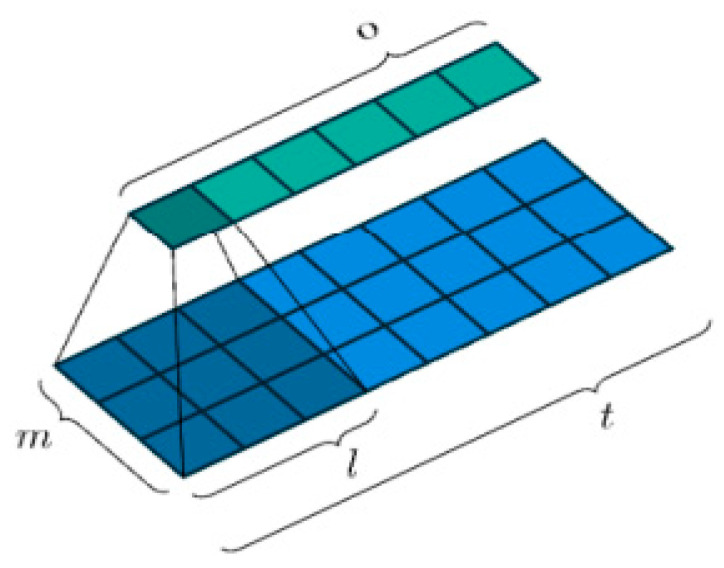
Defining a trainable weight matrix for the TDNN [85].

**Figure 12 sensors-24-02796-f012:**
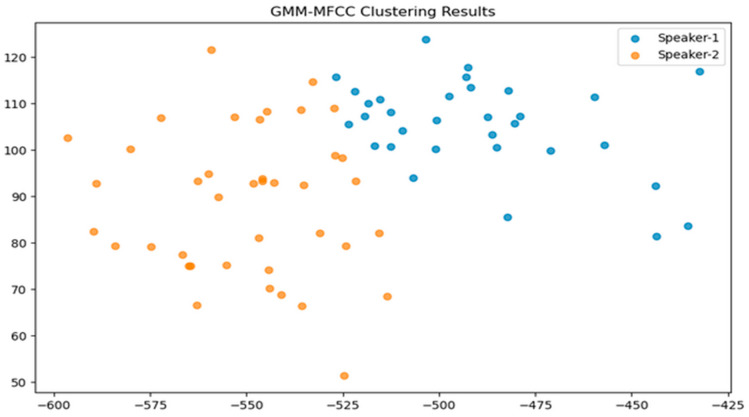
Scatter plot illustrating the clustering outcome of GMM-MFCC.

**Figure 13 sensors-24-02796-f013:**
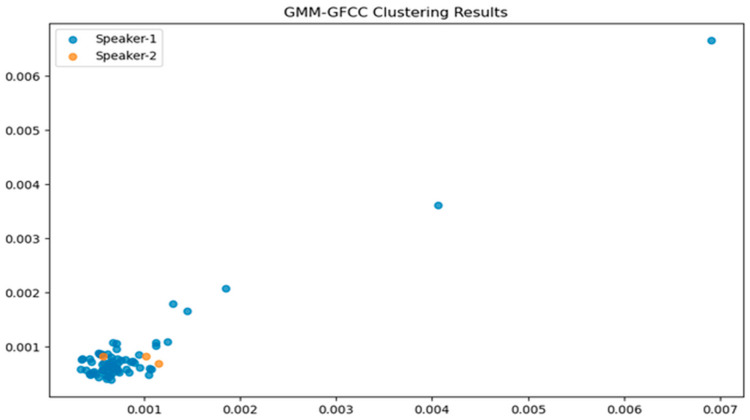
Scatter plot illustrating the clustering outcome of GMM-GFCC.

**Figure 14 sensors-24-02796-f014:**
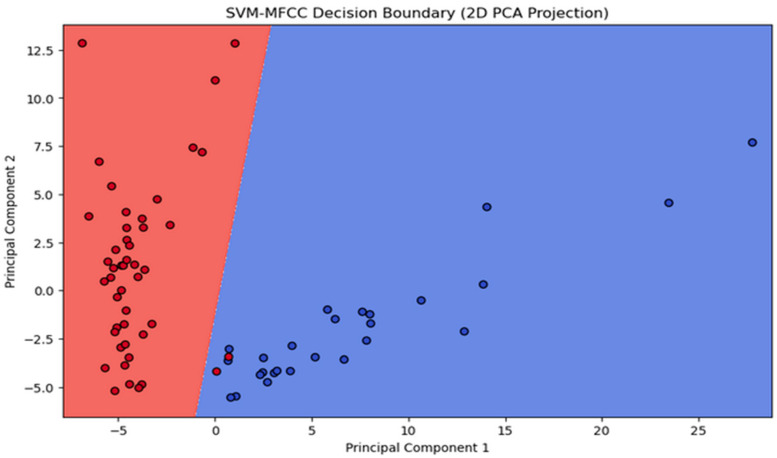
Scatter plot illustrating the clustering outcome of SVM-MFCC.

**Figure 15 sensors-24-02796-f015:**
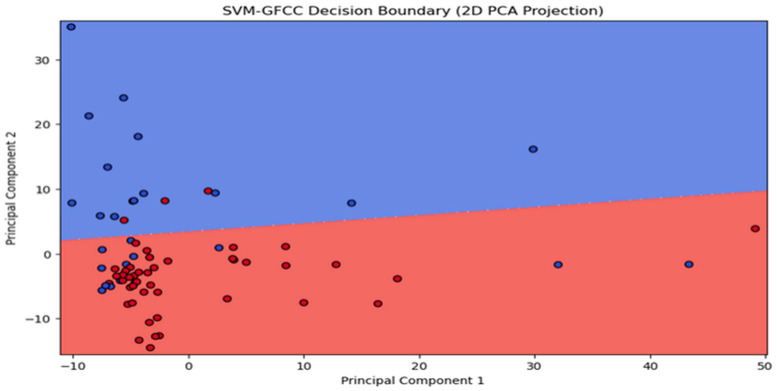
Scatter plot illustrating the clustering outcome of SVM-GFCC.

**Figure 16 sensors-24-02796-f016:**
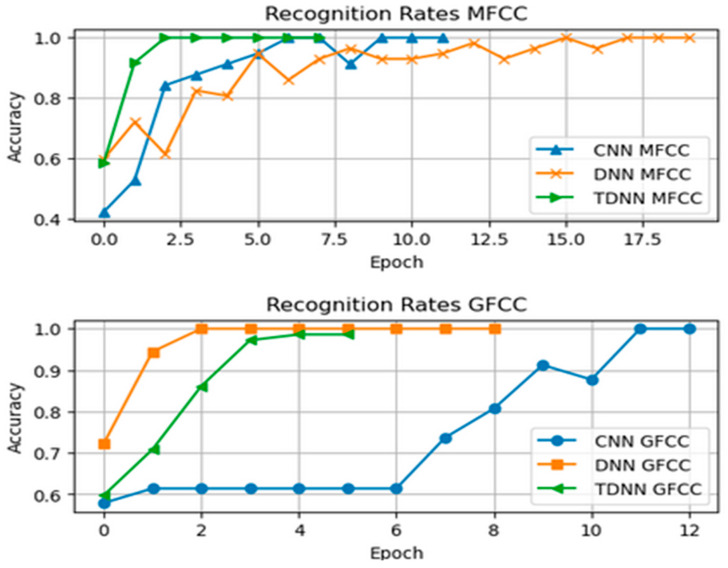
Recognition Rates of the Neural Networks using MFCC and GFCC combined features.

**Figure 17 sensors-24-02796-f017:**
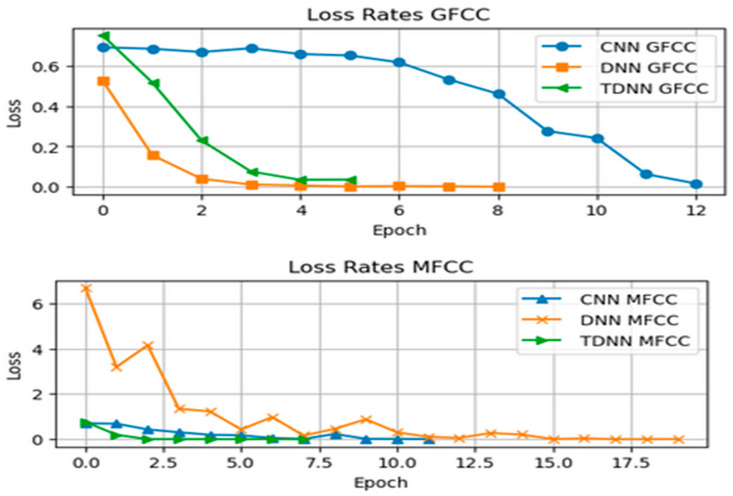
Loss Rates of the Neural Networks using MFCC and GFCC combined features.

**Figure 18 sensors-24-02796-f018:**
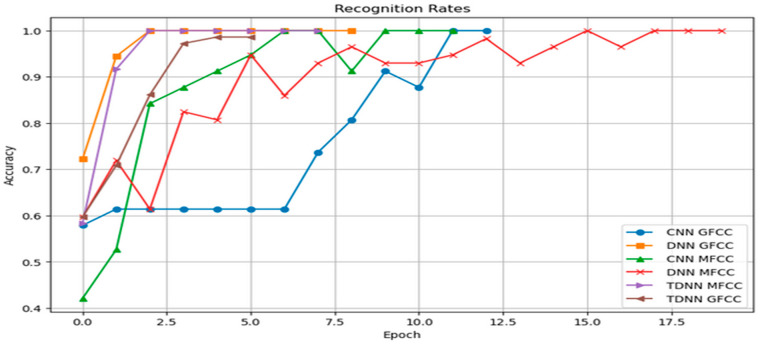
Recognition Rates of Neural Network Models for Active Speaker Classification.

**Figure 19 sensors-24-02796-f019:**
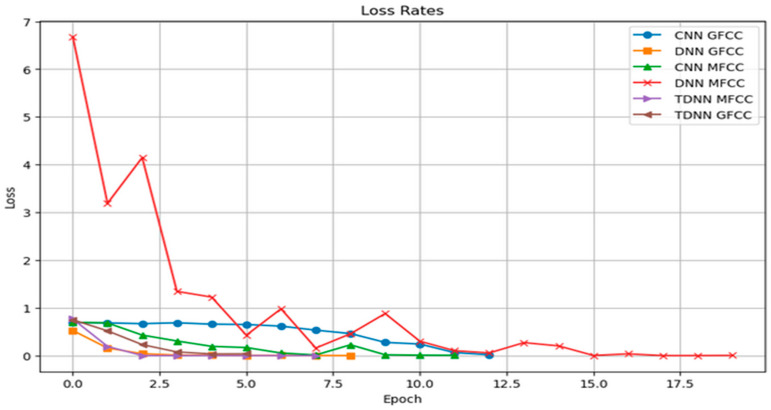
Loss Rates of Neural Network Models for Active Speaker Classification.

**Figure 20 sensors-24-02796-f020:**
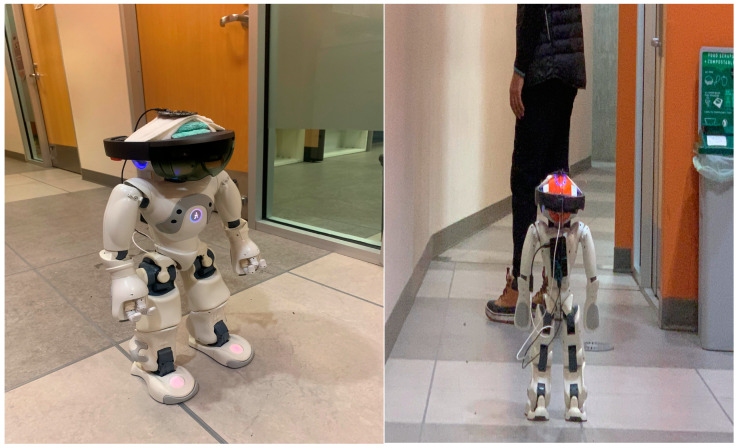
The integrated system into the Nao robot head alongside Microsoft’s HoloLens and the microphone array.

**Figure 21 sensors-24-02796-f021:**
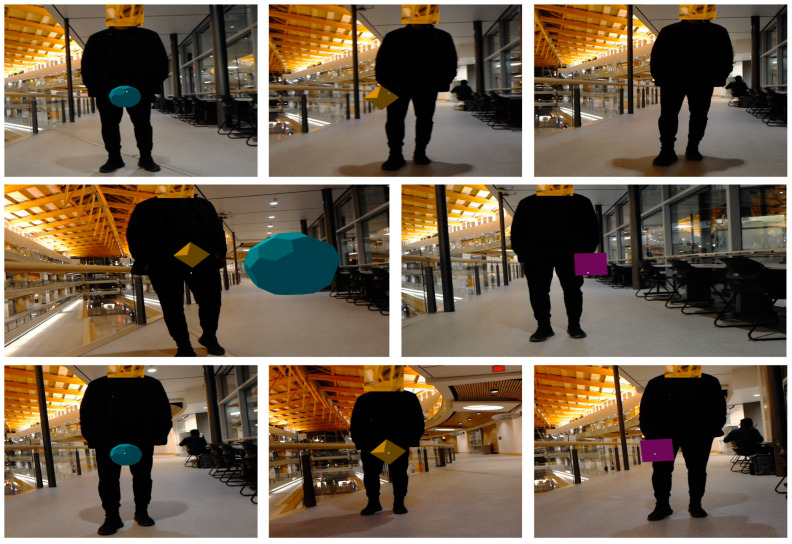
The audio-based virtual HoloSLAM robot environments.

**Figure 22 sensors-24-02796-f022:**
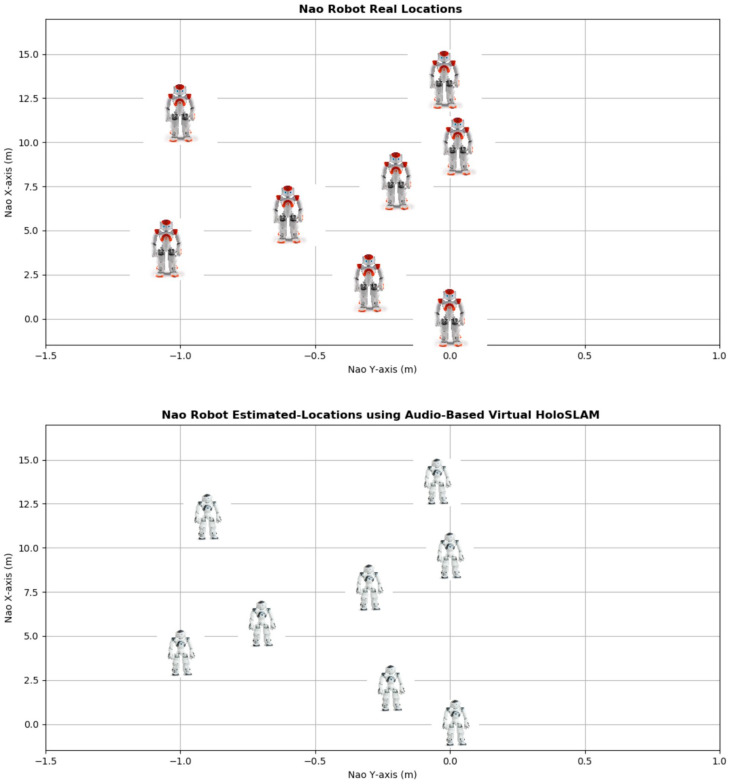
Estimated Nao robot locations using audio-based HoloSLAM.

**Figure 23 sensors-24-02796-f023:**
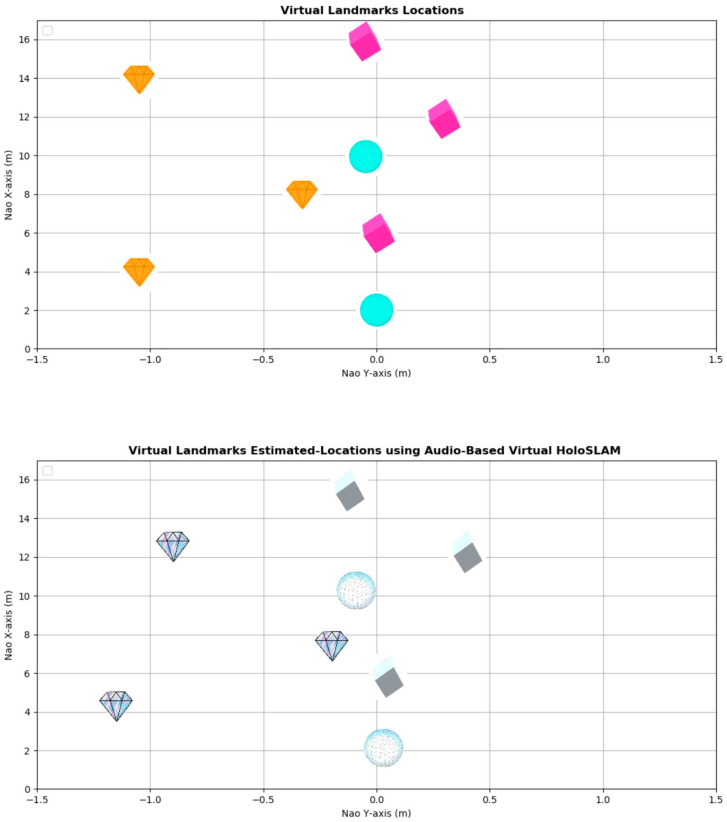
Estimated virtual landmark location using audio-based HoloSLAM.

**Table 1 sensors-24-02796-t001:** Comparative Analysis of Gaussian Mixture Model Covariance Types on K-Fold Split MFCC and GFCC Data.

	Covariance Type							
k-th Split	MFCC				GFCC			
k	Full	Diag	Tied	Spherical	Full	Diag	Tied	Spherical
1	79.687500	43.750000	35.937500	32.031250	39.062500	50.000000	44.791667	43.359375
2	21.875000	14.843750	17.187500	18.359375	39.062500	39.062500	39.062500	40.234375
3	78.461538	42.307692	55.384615	46.153846	38.461538	36.153846	36.923077	43.846154
4	21.538462	13.846154	35.384615	31.923077	38.461538	50.000000	46.153846	43.076923
5	21.538462	56.923077	45.641026	39.615385	44.615385	41.538462	40.512821	40.384615
6	20.000000	56.923077	44.615385	53.076923	32.307692	35.384615	35.897436	36.153846
7	84.615385	87.692308	86.666667	68.846154	32.307692	33.846154	32.820513	33.461538
8	84.615385	46.923077	57.435897	46.923077	40.000000	37.692308	47.179487	45.000000
9	83.076923	44.615385	36.410256	47.307692	40.000000	50.000000	54.871795	50.000000
10	23.076923	14.615385	36.410256	47.307692	69.230769	53.846154	48.717949	45.769231

**Table 2 sensors-24-02796-t002:** Summary of results of all six models trained across their testing Recognition Rates and Loss Function Values along with the features used during preprocessing.

	Training Parameters	Recognition Rate (in %)	Loss Function Value	Combined Features Used (MFCC/GFCC)
Conv-LSTM	824,322	93.75	0.1989	MFCC
Dense Neural Network	429,936	87.5	0.6421	MFCC
**Time Delay Neural Network**	**88,322**	**100**	**0.0003**	**MFCC**
Conv-LSTM	824,322	90.625	0.2838	GFCC
Dense Neural Network	429,936	93.75	0.1724	GFCC
**Time Delay Neural Network**	**88,322**	**93.75**	**0.1666**	**GFCC**

Optimizer = Adam; Learning rate = 0.0001; Loss = Categorical cross entropy; Metrics = Accuracy; Training iterations = 20.

**Table 3 sensors-24-02796-t003:** Nao robot parameters.

Specification	Details
Height	58 cm (22.8 inches)
Weight	4.3 kg (9.5 lbs.)
Degrees of Freedom	25
Sensors	-two HD cameras
	-four microphones
	-Touch sensors (head, hands, feet)
	-Inertial measurement unit (IMU)
	-Ultrasonic sensors
Processing Unit	Intel Atom Z530 processor
Memory	1 GB RAM
Operating System	Linux-based NAOqi OS
Connectivity	-Ethernet
	-Wi-Fi
	-Bluetooth
Power Source	Rechargeable lithium-ion battery
Battery Life	Up to 90 min of continuous operation
Development Framework	Choregraphe (graphical programming software)
	Python SDK
	C++ SDK

**Table 4 sensors-24-02796-t004:** Analyzing the performance of implemented audio-based virtual HoloSLAM.

Algorithm	Nao Position Error/m	Nao Orientation Error/rad	Virtual Landmarks Error/m
Nao IMU	33.01	0.675	
Audio-based Ellipsoidal Virtual HoloSLAM	0.0184	0.119	0.010
Total Times of identification speaker called	23		

## Data Availability

Data will be available upon request.

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
