# Peer review of "An Audio-Based SLAM for Indoor Environments: A Robotic Mixed Reality Presentation"

_sensors, 2024, doi:10.3390/s24092796_

Round 1

Reviewer 1 Report

Comments and Suggestions for Authors

This paper presents an audio-based SLAM using virtual landmarks.

There are several critical points to be improved:

- The main contributions are not clear. This paper highlights the benefits of combining the audio-based SLAM with the virtual landmark-based SLAM (HoloSLAM). However, most parts of this paper explain and evaluate how to extract voice features and match them (I am not sure how novel and significant they are). Thus, it is doubtful how significant such a combination is.

- Even though the details of HoloSLAM are given in [63], in the context of this paper, combining the virtual landmarks is not sufficiently explained with details (and even it is hard to guess them) For example, the explanation, page 26: 840-844 ("In this experiment, ) with Figure 20. The experiment is also under simple conditions. 

Minor:

- "Mixed Reality (MR) presents a novel approach to address the constraints of traditional landmark-based SLAM systems and associated landmark-related challenges [37]." : I do not understand the context of this statement. 

- [ citation for holoslam paper] --> [63]

- [49] "AIM-1120.pdf" --> with more information (e.g., link)

- Highlighting more relevant references would be helpful rather than listing existing numerous ones.

Comments on the Quality of English Language

Minors

Author Response

Dear  Reviewer,

We extend our heartfelt appreciation for your diligent review and insightful feedback on our submitted manuscript titled "An Audio-Based SLAM for Indoor Environments: A Robotic Mixed Reality Presentation," currently under consideration for publication in the Sensors MDPI journal. Your efforts in evaluating our work are truly valued.

We have carefully addressed each of the comments provided by the reviewers, offering comprehensive responses and implementing corresponding revisions to the manuscript. These adjustments have been meticulously integrated and are clearly indicated throughout the document, directly addressing the concerns raised during the review process.

We are confident that these revisions have significantly improved the quality and lucidity of the manuscript. We are grateful for the constructive criticism received, as it has undoubtedly contributed to refining the content and bolstering the overall presentation.

We eagerly anticipate further guidance and feedback from the editorial team. Your expertise is indispensable in ensuring the scholarly integrity and relevance of our research.

Warm regards,

Elfituri Lahemer and Ahmad Rad

Simon Fraser University

Reviewer 2 Report

Comments and Suggestions for Authors

Some figures should be improved with good description. The text should be outlined with related figures. Figure 6, should be outlined with Figure 6 (a) and (b) with good description. Figures 15-16 should be indicated with a) and b). Some technical specifications of the NAo robot should be given with table. Table 3 should be indicated with traning parameters of the neural network types. 

Comments on the Quality of English Language

English should be outlined with technical writing 

Author Response

(The authors gave the same response as above.)

Reviewer 3 Report

Comments and Suggestions for Authors

The authors present an interesting study regarding a novel location and mapping approach which utilises a sound source and microphone arrays to estimate the direction of a speaker.

The novelty of this manuscript is based on incorporating a robotic Mixed Reality using Microsoft HoloLens device to map the physical environment and to superimpose virtual landmarks on the location of the sound source called "HoloSLAM". This audio based method presents a novel approach that has several potential applications in the field of robotics and autonomous systems. Moreover, the authors provided a scientific suitable justification for their method. 

This reviewer considers that with minor adjustments this manuscript can contribute to advancements in autonomous robot navigation, human-robot interaction, and mapping technologies. Please follow the observations from bellow:

- Please define acronyms at first use (e.g. line 190).

- Line 287 “[citation for holoslam paper]” probably the authors wanted to cite an article.

- It is not clear if the authors are using the Hololens visual tracking system to improve the NEO’s robot accuracy. Please clarify.

- At lines 324-235 the authors stated that “Unity3D, Morphi, 3D Slash, Fusion 360, and Blender are valuable resources that provide a vast collection of pre-existing 2D/3D objects”  Some of these software are used purely to develop 3D models (e.g. Fusion 360) not as a database of 3D entities. Please clarify if all the mentioned software are used and if so, please offer more details for each one of them.

- The discussion section of the paper should be enhanced and contain a comparison of the accuracy obtained by the HoloSLAM method. 

- Even if it is presented in figures 21 and 22 the estimated robot location needs to be further discussed in therms of precision and at least compare to the robot’s standard navigation system. 

Author Response

(The authors gave the same response as above.)

Round 2

Reviewer 1 Report

Comments and Suggestions for Authors

The responses are mostly understandable.

Simply put, it seems that the main idea of this work is to address the challenge of audio-based SLAM in indoor environments by utilizing virtual landmarks incorporated by HoloSLAM [62].

While this integration is an interesting approach, it is confusing whether the main challenge (an autonomous robot with only a single microphone array and no cameras in indoor environments) is the valid assumption in environments where HoloSLAM [62] is available (HoloSLAM uses cameras and other sensors).

Furthermore, I anticipate that there are technical contributions in fusion-related parts, but they appear to rely on HoloSLAM [62] and the HoloLens framework. I guess that's why the article focuses more on the feature extraction part.

Please clarify these points.

minors:

- Please write explanations concisely (too lengthy).

- Please proofread the writing carefully again.

Comments on the Quality of English Language

Minors

Author Response

Dear Reviewer,

The authors appreciate very much the comments by the respected reviewer. We have studied those remarks carefully and have addressed all the points raised. We have also addressed all the above points (ticked boxes) in the revised version of the manuscript (highlighted sections). In this document, we specifically respond to reviewer comments.

Regards,
